# Prior mating success can affect allocation towards future sexual signaling in crickets

Rachel Chiswell[1], Madeline Girard[2], Claudia Fricke[3] and Michael M. Kasumovic[1]

[1] Evolution & Ecology Research Centre, School of Biological, Earth & Environmental Sciences, The University of New South Wales, Kensington, Sydney, NSW, Australia
[2] Department of Environmental Science, Policy and Management, University of California, Berkeley, CA, USA
[3] Institute for Evolution and Biodiversity, University of Muenster, Muenster, Germany

## ABSTRACT

Fitness is often correlated with the expression level of a sexually selected trait. However, sexually selected traits are costly to express such that investment in their expression should be optimised to maximize their overall fitness gains. Social interactions, in the form of successful and unsuccessful matings, may offer males one type of feedback allowing them to gauge how to allocate their resources towards sexual signaling. Here we tested whether adult male black field crickets (*Teleogryllus commodus*) modify the extent of their calling effort (the sexually selected trait) in response to successful and unsuccessful matings with females. To examine the effect that mating interactions with females have on investment into sexual signaling, we monitored male calling effort after maturation and then provided males with a female at two points within their life, manipulating whether or not males were able to successfully mate each time. Our results demonstrate that males alter their investment towards sexual signaling in response to successful matings, but only if the experience occurs early in their life. Males that mated early decreased their calling effort sooner than males that were denied a mating. Our results demonstrate that social feedback in the form of successful and unsuccessful matings has the potential to alter the effort a male places towards sexual signaling.

## INTRODUCTION

In many organisms, sexually selected traits are costly to produce and their expression often varies as a consequence of resource abundance (i.e., condition dependence; *Rowe & Houle, 1996*; *Cotton, Fowler & Pomiankowski, 2004*) (e.g., morphological traits: *Bonduriansky & Rowe, 2003*; ejaculate traits: *Fricke, Bretman & Chapman, 2008*; behavioural traits: *Lomborg & Toft, 2009*). This is most strongly reinforced by studies demonstrating longevity costs of extra investment in sexual signalling (*Hunt et al., 2004*), although questioned by studies that demonstrate the opposite (*Kotiaho et al., 1999*; *Papadopoulos et al., 2004*). Despite

Corresponding author
Michael M. Kasumovic,
m.kasumovic@unsw.edu.au

the well-established link between nutrient/resource abundance and trait expression (*Bonduriansky, 2007*), resources are only one of the many factors that influence the expression of sexually selected traits. Decades of competition research suggests that the social environment is an important contributor that results in plasticity in the expression of sexually selected traits after maturity. For example, in many species of mammals, birds, and invertebrates, dominant individuals are able to moderate the behaviour of subordinate individuals by enforcing a lower level of sexual signalling (*Qvarnström, 1997*; *Bekoff & Dugatkin, 2000*; *Tibbetts & Dale, 2004*).

As studies explore the effect of intra- and intersexual social interactions on individual behaviour, a secondary view that is gaining some traction in the literature is that plasticity in sexual signalling is a result of individual decisions resulting from conspecific feedback (i.e., information received from interacting with and eavesdropping on conspecifics) rather than enforcement. For example, studies manipulating the social environment during immaturity demonstrate that males of a wide variety of species alter how they allocate resources towards life-history (*Gage, 1995*; *Kasumovic & Andrade, 2006*; *Walling et al., 2007*; *Kasumovic et al., 2011*), metric (*Rodd, Reznick & Sokolowski, 1997*; *Kasumovic & Andrade, 2006*; *Kasumovic et al., 2011*; *DiRenzo, Pruitt & Hedrick, 2012*), physiological (*Stolz, Andrade & Kasumovic, 2012*), sperm (*Gray & Simmons, 2013*), and behavioural traits (*Bailey, Gray & Zuk, 2010*; *Kasumovic et al., 2012*) at maturity. Additionally, adult males show plastic responses when exposed to rivals, enabling them to quickly respond to changes in the environment (*Bretman, Gage & Chapman, 2011*) often with significant fitness benefits (*Bretman, Fricke & Chapman, 2009*). The feedback provided from the composition of the social environment can thus play an important role in individual perception of the intensity of competition, and therefore, the expression of sexually selected traits (*Kasumovic & Brooks, 2011*).

The idea that changes in sexual signalling are a result of social feedback rather than social enforcement is strengthened by the theoretical and empirical understanding of winner and loser effects in the competition literature. Repeated competitions with rivals for access to resources provides individuals with feedback on their own performance relative to rivals within a population (*Hsu, Earley & Wolf, 2006*; *Fawcett & Johnstone, 2010*; *Fuxjager & Marler, 2010*; *Kasumovic et al., 2010*). Individuals that win more contests are more likely to escalate future contests because they have a favourable perception of their fighting ability, while individuals that lose more contests have a less-favourable perception of their own fighting ability (i.e., winner/loser effects; *Hsu, Earley & Wolf, 2006*). In this sense, individuals are altering their behaviour and the expression of a sexually selected trait not as a consequence of condition, *per se*, but rather as a result of social interactions that indicate their relative fighting ability.

*Fawcett & Bleay (2009)* theoretically extended the idea of intra-sexual interactions modifying self-perception of fighting ability to inter-sexual interactions altering a male's perception of his own attractiveness. Through their model, they demonstrate that experiences of rejection reduce choosiness and experiences of acceptance increase it, similar to winner and loser effects in direct male-male competitions (e.g., *Kasumovic et al., 2009*).

Empirical evidence of mate rejection and acceptance in zebra finches (*Taeniopygia guttata*) supports this idea (*Pariser, Mariette & Griffith, 2010*; *Royle & Pike, 2010*). Moreover, when *Pariser, Mariette & Griffith (2010)* manipulated individual attractiveness through coloured leg rings, they found that zebra finch males subsequently changed body mass, condition and courtship display rate, demonstrating the importance of social feedback in trait maintenance. These results illustrate that artificial ornamentation not only manipulated the female's perception of the male but also changed the male's perception of his own attractiveness, via the behavioural 'mirror' of female feedback. Whether the change in courtship rates (i.e., sexual signalling effort) was a result of direct social feedback or a consequence of reduced mass and condition, however, is difficult to determine.

Our goal in this study was to examine whether feedback from females in the form of successful or unsuccessful matings alters future male investment in sexually selected signals. We used the Australian black field cricket (*Teleogryllus commodus*) to investigate this question as males call throughout their adult lifetime in an attempt to attract mates. Males that call more attract significantly more mates (*Bentsen et al., 2006*), but calling is likely energetically costly (e.g., *Hoback & Wagner, 1997* on *Gryllus lineaticeps*) as it leads to a shorter lifespan (*Hunt et al., 2004*; *Hunt et al., 2006*); thereby demonstrating the trade-off between reproductive effort and lifespan. In addition, there is evidence in this and other closely related species that social interactions and feedback affect juvenile developmental tactics (*Bailey, Gray & Zuk, 2010*; *Kasumovic et al., 2011*; *DiRenzo, Pruitt & Hedrick, 2012*; *Kasumovic et al., 2012*), adult male behaviour (*Judge et al., 2010*), and lifetime calling effort (*Zajitschek et al., 2012*; *Callandera et al., 2013*). As a result, we predicted that males would differentially alter investment towards sexual signalling as a function of successful mating experiences. We also wanted to explore whether we could partition the effects of mating itself and interactions with females to better understand how males invest in sexual signalling.

We hypothesized that successful mating opportunities would result in positive feedback that would reinforce the current signalling effort as being sufficient to attract and successfully mate with females. In contrast, we hypothesized that unsuccessful mating opportunities would provide negative feedback on reproductive effort and further suggest a choosier mating environment which should result in males increasing their calling effort. Since calling effort is costly, we predicted that these changes in calling effort should have subsequent effects on lifespan.

## METHODS

The crickets used for this experiment were first generation descendants of approximately 300 immature individuals collected at Smith's Lake, New South Wales (32°22′S, 152°30′E) in 2012. We removed penultimate male and female crickets from the common population and reared them individually in a plastic container (5 × 5 × 3 cm) with an egg carton for shelter, and *ad libitum* food (Friskies Go Cat Senior) and water. We kept all the individuals in a controlled temperature room of 28 °C on a 14:10 light to dark cycle and replaced their water and food weekly. We checked crickets daily for adult eclosion and within 24 h

of eclosing, we weighed individuals using an electronic balance and measured pronotum width as a measurement of size. Adult females remained in individual containers within a controlled temperature room until required for matings.

Upon maturity, we placed males in a custom-built electronic monitoring device (callbox; see *Lailvaux, Hall & Brooks, 2010*) overnight each day to determine daily calling effort. Males were kept in individual containers ($5 \times 5 \times 3$ cm), which were then placed in plastic containers ($14 \times 6 \times 6$ cm) surrounded by acoustic foam to keep males in acoustic isolation. We monitored male calling over a 12 h period each night and used a male's time spent calling each night in seconds as our estimate of calling effort. A male's placement within the callbox was randomized every other day.

Although males were placed within the callbox upon maturity, we only monitored male calling effort from age 7–28. On day 14 and 21, we removed males from their individual cages and placed them in a $10 \times 5 \times 4$ cm mating arena. As males generally begin calling between days 5–10, and usually reach their peak calling effort around day 20–22, we chose day 14 and 21 to examine how social interactions affect calling effort after males have begun calling to before reaching their peak. We chose two mating opportunities as males in a natural population of another species of cricket (*Gryllus campestris*) averaged two matings throughout their lifetime (*Rodriguez-Munoz et al., 2010*) and we wanted to examine whether single mating opportunities affected age-specific calling effort. Once in the mating enclosure, a virgin female was placed within the container for 24 h along with her water and food containers and males were either allowed to interact and mate with the female or allowed to interact but denied the opportunity to mate. Previous studies demonstrate that males always mate when placed with a female over 24 h (*Hall et al., 2010*; *Lailvaux et al., 2011*; *Zajitschek et al., 2012*). To ensure that selected males were denied access to females, we placed females inside an overturned 100 ml transparent plastic cup. Each cup had approximately twenty holes evenly distributed throughout the cup and no bigger than 4 mm in diameter. These holes were large enough to allow males to pass the front of their head and antenna through, but not their body. These holes thus allowed pheromonal, acoustic and visual communication to occur between the males and females, but prevented any matings from occurring.

As males were either granted or denied a mating at each mating opportunity (day 14 and 21), we had a $2 \times 2$ design which resulted total of four experimental treatments: (i) an 'early mating' treatment where males were granted a mating on day 14 but denied a mating on day 21, (ii) a 'late mating' treatment where males were granted a mating on day 21 but denied a mating on day 14, (iii) a 'multiple mating' treatment where males were granted a mating on both day 14 and 21, and (iv) a 'denied mating treatment' where males were prevented from mating on both day 14 and 21. We also had an isolation (control) treatment where males were placed in a mating arena but no females were introduced. Since no female was introduced, we placed the male's own egg carton and water bottle in the mating arena to ensure he had food and water. All the males were returned to the call box the day after their mating treatment and calling effort was monitored until day 28. After 28 days, we removed males from the call box and monitored their survival.

Although all previous studies demonstrate that male *T. commodus* mate immediately with females when they are available (*Hall et al., 2008*), there is the possibility that some males did not mate or that some males mated more often than others. This, however, would add noise to the differences in age-specific calling effort between treatments making it less likely to see a significant effect of a mating rather than skewing it towards significance.

## Statistical analysis

In all our examinations, we used a mixed-model approach with individual identity as a random factor using the nlme package (*Pinheiro et al., 2009*) in R 3.1.0 (*R Core Team, 2014*). We log transformed calling effort and used the orthogonal quadratics to avoid issues with multicollinearity. Since males began calling at different ages resulting in different rates of calling increase early in their lifetime, it resulted in significantly different early portions of the calling curve. As a result, we limited our analysis to calling effort between days 15 (the first mating) and 28 to ensure that calling effort prior to the treatment did not skew our results and that we focused on changes in calling effort as a function of the treatment.

As the individuals in the isolation treatment never experienced any females throughout their lifetime, there was no direct way to examine whether males in the isolation treatment behaved differently to all the mating treatments without classifying each treatment separately, thus decreasing our power. As such, we first examined whether age-specific calling effort of males in the isolation treatment differed from the males in the denied mating treatment by adding a treatment effect to a model containing only the linear and quadratic components of age. Since adding treatment to the model did not improve our fit (see results), we decided to use the isolation treatment by combining it with the denied treatment in our further analyses to maximize our sample size and power.

To examine whether early and late matings each individually or jointly affected age-specific calling effort, we coded each mating (early and late) as a function of being granted or denied. Since our null hypothesis was that each mating would affect future investment in calling effort, we tested a model that contained the linear and quadratic effects of age, the main-effects of early and late matings, as well as the two-way interactions between the early and late mating terms and both age terms (Table 1). We also placed male as a random effect in our model and report statistics using REML. We visualized male age-specific calling effort using non-parametric splines generated with the general additive mixed model package (gamm4, *Wood, 2009*).

We used an ANOVA to examine whether total calling effort (log transformed) differed as a consequence of either an early or late successful mating, and a GLM to examine whether lifespan was affected by a successful early or late mating with total calling effort as a covariate. These were performed in R 3.1.0.

## RESULTS

A total of 102 males were approximately equally distributed within each of the five treatments (Early: 22, Late: 22, Multiple: 19, Denied: 20, Isolation: 19). Each treatment began with 25 individuals and the difference in the sample size is a result of individuals that

**Table 1 The results of the model examining differences the effect of early mating on age-specific calling effort.**

| Factor | d.f. | F-value | p-value |
|---|---|---|---|
| Age | 1,1198 | 23.31 | <0.0001 |
| Age$^2$ | 1, 1200 | 7.86 | 0.005 |
| Early mating | 1, 99 | 0.01 | 0.74 |
| Late mating | 1, 99 | 0.18 | 0.67 |
| Early mating × Age | 1, 1198 | 2.29 | 0.13 |
| Late mating × Age | 1, 1198 | 0.11 | 0.74 |
| Early mating × Age$^2$ | 1, 1198 | 5.54 | 0.019 |
| Late mating × Age$^2$ | 1, 1198 | 1.23 | 0.27 |

**Table 2 Model regression coefficients.** The regression coefficients of the linear (age) and quadratic (age2) terms of age for the individuals denied and granted their first mating opportunity.

| | Early mating | Estimate ± SE | p-value |
|---|---|---|---|
| Linear | Denied | $-8.46 \pm 0.53$ | 0.006 |
| | Granted | $-15.90 \pm 0.72$ | <0.0001 |
| Quadratic | Denied | $-2.10 \pm 1.83$ | 0.49 |
| | Granted | $-13.43 \pm 7.07$ | <0.0001 |

didn't call at all within the first 14 days of their life being removed from the experiment. We removed these individuals since males that do not call by this point also did not call throughout the rest of their life, likely because they were following a different mating strategy (i.e., sneaking rather than calling; *Bailey, Gray & Zuk, 2010*). As a result, keeping these non-calling males would skew results in the different treatments. In addition, although individuals may alter other behaviour, we cannot study changes in calling effort if males do not call. No individuals died before 28 days. As one individual escaped from the Late mating treatment prior to death, the lifespan analysis consists of 101 individuals.

In our initial comparison of the age-specific calling curves between individuals in the isolation and denied treatment, the addition of treatment in the model did not significantly improve the fit (log-likelihood difference = 0.66, $P = 0.42$). As a result, we combined the two treatments to improve our sample size.

The model that best described the variation in age-specific calling effort included the early mating term, the linear and quadratic effects of age, as well as the interaction between the early mating term and the linear and quadratic effects of age (Table 1). This results in changes in the curvature of lifetime male calling effort with age (Fig. 1). In our model, only the interaction between the early mating and the quadratic term was significant (Table 1), suggesting that it is the non-linear component that is affected with an early mating. This results in changes in the curvature of lifetime male calling effort with age (Table 2, Fig. 1).

There was no difference in total calling effort as a function of a successful early ($F_{1,99} = 0.08$, $P = 0.78$) or late ($F_{1,99} = 0.00$, $P = 0.99$) mating between treatments. Individual

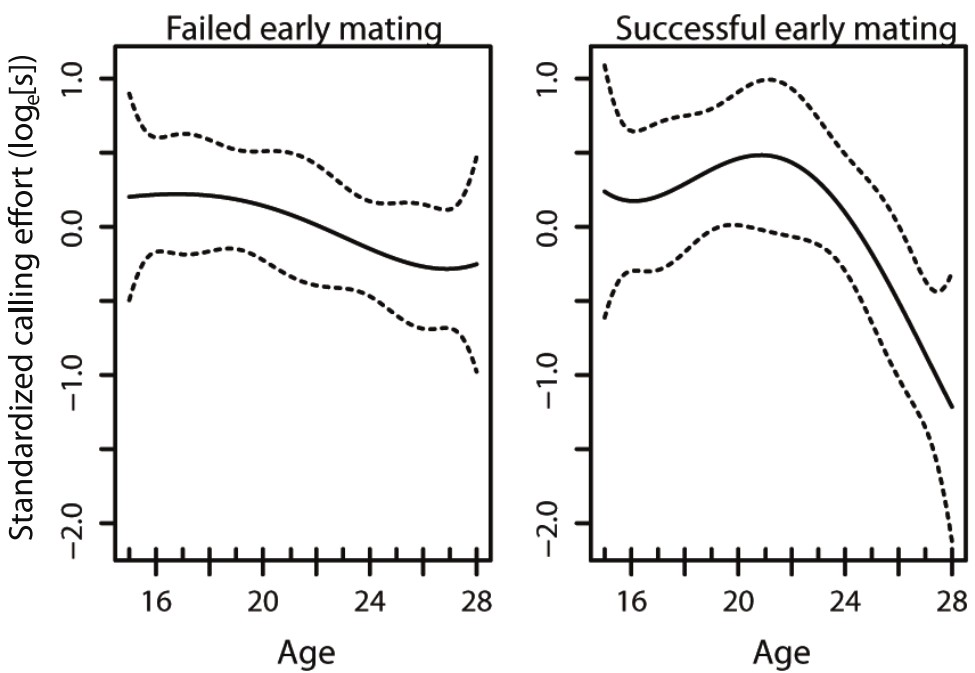

**Figure 1 The age-specific calling curves of males that failed and succeeded in their early mating.** The dotted lines are 95% confidence intervals.

lifespan also was not affected by either a successful early ($t_{1,97} = -0.0.5, P = 0.96$) or late ($t_{1,97} = 0.82, P = 0.41$) mating, or total calling effort ($t_{1,97} = 0.95, P = 0.34$).

## DISCUSSION

Given the strong selection for high rates of acoustic calling in *Teleogryllus commodus* (*Bentsen et al., 2006*), there is likely to be an evolutionary benefit for males to optimise calling effort over their lifespan. Both diet (*Hunt et al., 2004*; *Judge, Ting & Gwynne, 2008*; *Bertram et al., 2011*) and the social environment encountered prior to maturity (*Kasumovic, Hall & Brooks, 2012*; *Kasumovic et al., 2012*) contribute to the variance in male age-specific calling effort trajectories. Here we investigated whether variation in age-specific calling effort may also be a response to social feedback in the form of successful and unsuccessful matings. Our results suggest that social feedback from matings occurring early in a male's lifetime have the potential to alter how they invest in sexual signalling throughout their life (Fig. 1). Despite this shift in the trajectory of calling effort, there was no difference in the total calling effort of males in the different treatments and no effect on male lifespan.

A previous study examining the effect of diet and weekly successful matings in *T. commodus* found that males that were mated on a weekly basis had a lower lifetime calling effort than virgin males (*Zajitschek et al., 2012*). Our results broaden our understanding of sexual investment by males as we demonstrate that a shift in investment can occur as a consequence of a single successful mating. Our mating treatments were positioned such that they occured before a virgin male's peak calling effort in an attempt to understand whether

mating interactions affect how males reach their peak calling effort. We show that an unsuccessful mating prior to a male reaching his peak calling effort results in males maintaining their calling effort for a longer period of time relative to individuals that are successful early in their lifetime (Fig. 1). Our results thus provide evidence that males use social feedback from females to alter their investment towards sexual signalling as seen in other species (*Pariser, Mariette & Griffith, 2010*; *Royle & Pike, 2010*; e.g., *Callandera et al., 2013*). Most importantly, we demonstrate that males that mate early in their lifetime subsequently differ in their signalling effort compared to males that do not mate early. As a result, no social feedback from conspecific males in the form of social enforcement is necessary (*Callandera et al., 2013*) for individuals to alter their investment towards sexual signalling.

Another possibility is, however, that mating itself was costly enough that it decreased male investment in sexual signalling. For example, in tree crickets (*Oecanthus nigricornis*), males provide a nuptial gift and what were considered high quality males decreased their investment in subsequent gifts in response to perceived female density (*Bussière, Basit & Gwynne, 2005*). Although *T. commodus* do not produce gifts as large as *O. nigricornis*, the spermatophores produced prior to courtship may be costly enough to result in a decrease in calling effort. Our mating design allows us to further dissect this possibility. First, regardless of whether males successfully mated or not, all male *T. commodus* perform the costly courtship calling (*Hack, 1998*) prior to mating and produce a spermatophore in preparation for mating (*Loher, 1974*). This coupled with the fact that there was no difference in lifespan as a function of a successful mating suggests that similar contact costs were paid by all males in the mating treatments. Second, we did not see a change in age-specific calling effort as a function of a granted or denied second mating, further suggesting that decreased calling effort is not a consequence of a decreased energetic pool. Finally, males in the isolation (control) treatment where the extra-costly courtship calling and spermatophore transfer was presumably not triggered did not significantly differ in age-specific calling effort from males in the denied treatment. These factors suggest that a change in age-specific calling effort is a function of female feedback rather than spermatophore or mating costs, but further studies are necessary to rule this possibility out completely.

Despite the differences in age-specific calling effort between treatments, we saw no difference in lifetime calling effort or male lifespan between males as a consequence of successful matings in either the early or late period. This is in contrast to previous results demonstrating that repeated matings reduce calling effort (*Zajitschek et al., 2012*). This difference may be because males were continually mated throughout their life, rather than provided with only two matings well-spaced through the early period of their life in our experiment. Nevertheless, it may be beneficial for males to alter their signalling effort in response to female feedback early in their life because a male's likelihood of survival under field conditions is significantly lower than a female's (15.2 vs. 21.4 days, respectively) (*Zajitschek et al., 2009*). Males that decrease their calling effort in response to their success may increase their natural lifespan by decreasing their mortality hazard rate from parasites and predators that exploit sexual signals (*Zuk & Kolluru, 1998*). This idea, however, requires further testing in a field scenario.

Our results add to the growing list of studies demonstrating that social feedback can influence individual reproductive effort in a myriad of ways ranging from experiencing them prior to maturity (*Kasumovic & Brooks, 2011*), to direct (*Fawcett & Johnstone, 2003*; *Hsu, Earley & Wolf, 2006*) and indirect (*Callandera et al., 2013*) intrasexual feedback, and direct intersexual feedback (*Fawcett & Bleay, 2009*; *Pariser, Mariette & Griffith, 2010*; *Royle & Pike, 2010*). The experience an individual gains as a function of conspecific interactions is thus an important determinant of the level of trait expression, selection and therefore, how researchers view the concept of individual quality (*Lailvaux & Kasumovic, 2011*). Our results, in combination with others, highlight the importance of examining and considering the social environment and social experience when exploring male mating strategies and investment towards sexual signalling.

## ACKNOWLEDGEMENTS

Thanks to Heather Try for help in rearing crickets, Rob Brooks for looking at earlier drafts in the manuscript, and a huge shout out to Matt Hall and James Smith for discussions about statistics. We would also like to thank Nathan Bailey, two other anonymous reviewers and especially Luc Bussière, for helpful comments during the review process. MBG was supported by NSF's East Asia and Pacific Summer Institute program.

### Funding

CF was supported by travel grants from Helge Ax:son Johnsons Stiftelse and Stiftelse Lars Hiertas Minne. MMK and the research was supported by an Australian Research Council DECRA Fellowship. The funders had no role in study design, data collection and analysis, decision to publish, or preparation of the manuscript.

### Grant Disclosures

The following grant information was disclosed by the authors:
Helge Ax:son Johnsons Stiftelse and Stiftelse Lars HiertasMinne.
Australian Research Council DECRA Fellowship.

### Competing Interests

The authors declare there are no competing interests.

### Author Contributions

- Rachel Chiswell conceived and designed the experiments, performed the experiments, wrote the paper, reviewed drafts of the paper.
- Madeline Girard performed the experiments, reviewed drafts of the paper.
- Claudia Fricke conceived and designed the experiments, performed the experiments, reviewed drafts of the paper.

**Peer**J

- Michael M. Kasumovic conceived and designed the experiments, performed the experiments, analyzed the data, contributed reagents/materials/analysis tools, wrote the paper, prepared figures and/or tables, reviewed drafts of the paper.

### Supplemental Information

Supplemental information for this article can be found online at http://dx.doi.org/10.7717/peerj.657#supplemental-information.

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
