# Peer review of "Prior mating success can affect allocation towards future sexual signaling in crickets"

_PeerJ, doi:10.7717/peerj.657_

## Round 0.1 · original submission · Major Revisions

Overall, I found this manuscript to be a very strong study and one that highlights the importance the social environment can play in understanding male mating strategies and how males invest differently in their sexual signalling. This was a fun read!

However, the two reviewers had some valid concerns. I would like you to carefully consider both reviewers comments, especially Reviewer 1's concern with the time-scale of the study and Reviewer 2's concern with excluding non-signalling males from the study. Both reviewers points are valid. If the time scale is shifted to one that more appropriately represents the lifespan of the individuals in the field, how much do the results changes? Further, if all individuals are included (instead of excluding the non-callers) how much do the results change? I think these two ideas need to be more fully explored and will want to see this in a revised version.

Reviewer 1 ·

Basic reporting

On the whole the article is well-written and provides sufficient background to frame the study. The tables and figure need a little work to be publication ready. The formatting of the tables makes them difficult to read because entries spill on to multiple lines. Please adjust the spacing within and between cells, and the cell widths to make the tables easier to read.

I would like to see an axis label on the y-axis of Figure 1, if possible, or reference to the appropriate section of text that explains that measure of signalling effort. It is not an intuitive axis and thus the figure cannot be understood without reference to the Methods and/or Results, which is also not included. The unit of measurement given in the Methods is time spent calling per night, and so I assume these are somehow normalized or transformed values of that measure? Please clarify. Additionally, this figure is not currently referenced in the Results, so I’d suggest adding a sentence there explaining what the figure displays and referencing the figure.

Experimental design

The experimental design seems appropriate for the interesting research question posed in the paper.

I do have a concern about the timescale considered in the study (explained in detail below) and I suggest that the authors consider including at least one night of calling prior to the first experimental mating treatment in their analysis, as well as consider not including the last bit of the experiment which is beyond the expected lifespan in the field. Perhaps an added exploratory analysis, looking at just days 15-21, or 15-24 even, could be included. I suspect the results may change if the last few days are taken out, as it looks like they are driving the significant interaction between age and treatment. At the very least, the authors need to clearly interpret the results and be more careful with how they discuss the effect of mating.

Validity of the findings

I am not confident with the way in which the results are discussed given the data. I accept some responsibility for that, not completely understanding the statistics (e.g., the quadratic effects of age), but I think that the authors need to re-visit what the statistics reveal and the way in which they discuss their results.

Looking at Figure 1, it looks like males who successfully mated at day 14 increased their calling rate, until the second trial on day 21 after which we see a sharp decline in calling effort. Conversely, males who did not mate maintained their level of calling for a few days and then gradually declined over time. It appears as though it is not until about day 24 that the males in both treatment groups display a similar level of signalling effort, after which the mated males decline at a higher rate. In your abstract, you state that “Males that mated early in their life decreased their calling effort sooner than males that were denied a mating.” While this is true for the later part of the experiment (beyond their typical lifespan in the field), they first increased their signalling effort after mating for a few days, and then decreased, while the unmated males maintained and then slowly decreased their effort. So I think that your summary does not really capture the whole story, and more discussion of what happens after that initial treatment for both groups is needed.

I also think that the main result that is discussed (the effect of mating, which itself is not significant in the model) needs to be clarified and discussed carefully. The authors found an interaction between the quadratic effect of age and the mating treatment. I could not find where they interpreted this result clearly, but instead seemed to focus primarily on the effect of mating. I think I likely have an average level of understanding of the stats used here relative to the target audience, and I am unclear as to what the quadratic effect of age x treatment means without much thought on my own part. I think the authors need to do a better job of making their main result clear for readers and stick to discussing what their results support. My interpretation of the results (based on Table 1 and Figure 1) is that mating treatment matters but only at older ages. Whether or not that effect is biologically relevant deserves to be discussed as well, in my opinion.

Additional comments

Line 10: The opening sentence of the introduction seems too strong, especially given that the next sentence mentions that some studies have found the opposite. Please change the wording appropriately (e.g., by adding “Many” or “In general,” at the start of the sentence and adding “often” to the second part of the sentence to result in “… their expression often varies…”, or something similar).

Line 26: This is the first time of many you use the term feedback (alongside other words like conspecific feedback, feedback from the composition of the social environment, …. ) I think it would be useful to explain what you mean by feedback. I understand feedback to be a response that influences future output in a system. But I found it a little confusing on my first read of the paper because of how it is used in the paper. Even just a brief definition and/or citation in parentheses after it is mentioned the first time would be a helpful improvement.

Line 28: I think it would be helpful for readers if you add more information about which males you are referencing in this statement. For example, you could simply add “… of many taxa…” after “males”, or in the parentheses with the citations add to which species or taxa the studies refer.

Line 45: “lower perception” is potentially confusing as it is not clear if you mean the losers have less perception or a less favourable perception of their abilities. I would recommend changing the wording from “lower” to “less favourable” or “negative” or something similar to be clear about your meaning.

Line 137: So you did not look at calling effort before the first mating and only after? I would think that a comparison between before any treatment (e.g. day 13) would be interesting to compare with directly after the first treatment (which occurs on day 14, so the first night of calling recorded would be on day 15, if I understand correctly). Your rationale for excluding calling that occurred before the first treatment is so that it did not skew the results. I can understand this for the first days of calling when there is a lot of variation between males, but weren’t all males who included in the experiment calling before the first treatment (as indicated by the note that you excluded males who did not call on line 165)? And if so, I think you should include at least one night of calling before any treatments were performed so that you have a baseline of calling activity for each male to then compare against the change in calling after the treatments.

Line 176: Combining the isolation and denied treatments doesn’t increase your sample size, but does increase your statistical power by reducing the number of treatment in the models, if I understand correctly. Please change the wording here to reflect that the combination of treatments was to improve power and not sample size.

Line 181: I believe “in” is a typo here and should be deleted (“…quadratic term in for males…”).

Line 198: Can you state for certain that there was no effect on lifespan as a direct result of the lack of statistical differences in calling effort between treatments? I think it would be wise to add a qualifier here (e.g., “… different treatments, and likely as a result, no effect…”) or change the wording to delete the strong cause/effect link you currently include (i.e., simply delete “, and as a result,”).

Line 207: This may be true, but it leaves out that mated males actually increase their effort for a few days prior to then decreasing it. So I think there is an issue with the timescale on which you report your overall findings and I’d like it to be clearer what you are discussing here, and elsewhere.

Line 212: I’m not sure that you can claim that an early mating causes changes in signalling effort because you did not include a comparison of signalling effort before the mating. You show differences between mating treatments, so I think a more appropriate way to word this would be that males that mate early in their lifetime subsequently differ in their signalling effort compared to males that do not mate early.

Line 232: Again, I do not agree with calling your results a “shift” or a change in calling effort when what they appear to be is a difference in calling effort between treatments. Using the terms shift and change imply that you are comparing before and after a treatment, which I would recommend, but you aren’t currently. This sort of wording implies a comparison within groups of males (which it is not, if I understand correctly) as opposed to between groups of males. I would urge caution with this type of wording not merely from a semantics point of view but also because this is the type of statement that both the media and other scientists latch on to when reporting on studies, often without a complete understanding of the study design. I may be misunderstanding the statistical analysis, but in either case I think more clarity and precision when discussing results would be beneficial.

Line 239: Here you talk at once about mated males decreasing calling effort (which from Fig 1 looks like it happens from day 21 onwards) and the fact that males have a shorter lifespan in the field (15 days). I can’t help but wonder about the biological relevance of your reported decline in calling effort later in life, well past the typical lifespan in the field. How much of the decline in the last 7 days reported, which occurs for both mated and unmated males, likely has to do with them being old as opposed to whether they are mated or not? I think that the key result that is reported and discussed needs to be better backed up by both the stats and the biology of the organism to make a convincing case.

Line 244: This two sentence paragraph needs a bit more explanation to make it stand-alone and clear. My reading of it is that you are asserting that had you measured whether and how many times males mated during the trials and included that in the analysis you would be less likely to see an effect of mating treatment on calling effort, and therefore it is not something that you are worried about as a source of error.

Line 249: I found the concluding paragraph confusing given what was examined in the current study. What do you mean by “The competitive context is thus an important determinant…” and how does that relate to your study on the effect of mating on signalling effort? Possibly inserting a word or a few words here would help clarify your meaning (e.g., “The competitive intra- and intersexual context…”).

Line 257: While I agree that the social environment should be considered in studies of mating and signalling, I’m not sure there is a direct link to your study the way this sentence is currently written. Adding “and social experience” to the sentence such that it reads “… social environment and social experience when exploring…” would make that connection and would strengthen the final sentence of your paper.

Table 1 caption: I believe the word “in” is missing in this sentence (“… examining differences the effect of…”) and should be added.

Reviewer 2 ·

Basic reporting

No problems here.

Experimental design

The experimental design seems fine.

Validity of the findings

Are the data being archived in a public repository? This should be made clear.

Additional comments

I previously reviewed the paper, “Prior mating success can affect allocation towards future sexual signalling in crickets” by Chiswell et al. for another journal. It it the authors test for an effect of male crickets’ mating experience on their investment in calling effort. This is a worthy goal as male mating experience could modulate the strength of selection on male sexual signals as might be predicted by (among other things) senescence theory. To carry out the experiment, the authors create 5 treatments where they vary the number and timing of matings. In four of the treatments, males are either allowed free contact with a female for 24 hours (and a mating is assumed to have occurred) or presented with a female under a barrier where contact but no mating is allowed, and these conditions occur on day 14 or day 21 or both. The resulting treatments are: 1) multiple matings (allowed on both days 14 and 21), 2) early mating (on day 14, restricted on day 21), 3) late mating (on day 21, restricted on day 14), and 4) denied matings (restricted on both days 14 and 21). The final treatment, which the authors name a control, had no contact with females throughout the experiment. All males had their calling effort measured daily from day 15 to day 28.

Reading this version of the paper, the authors have made efforts to address my previous comments. However, they have not adequately addressed one comment that I believe substantially affects the interpretation and generalizability of the findings. Specifically, the authors removed a substantial subset of males in each treatment from the experiment (between 12 and 24% of the males depending on the treatment) because they did not call at all within the first 14 days of adult life (see lines 164-165). The authors’ rationale for removing these males just doesn’t make sense to me. These removed males could have potentially shown the greatest treatment effect. The fact that they were pursuing a different strategy hardly seems like a justification for excluding them. In addition, by virtue of their own behaviour these excluded individuals have opted themselves out of the experiment – it is never a good idea to let the experimental animals decide whether they are part of an analysis. To exclude such a large proportion of the sample size, and a proportion that could show very strong treatment effects on all response variables measured, seems like a very bad design flaw. Keeping the males in would not “skew” the results as the authors claim (line 169) but it would introduce error, which is an unavoidable feature of behaviour research. Also, the statement, “we cannot study changes in calling effort if males do not call” (lines 170-171) is logically false. The experiment manipulated experience and measured male calling effort over time and tested for differences in group means – there is no reason that all individuals have to alter their calling effort or call at all for the underlying hypothesis to be evaluated. These non-callers are part of the experiment and should contribute their behaviour to the results analyzed. It continues to be perplexing to me why the authors wouldn’t have just followed these low-effort callers through the experiment. In my opinion, this feature of the design makes it very difficult to interpret the results or to generalize them at all.

I really don’t like AIC approaches to analyzing data as they give over control of what variables are tested to the stats packages and the data in the sample, rather than being controlled by the researcher. Similar to stepwise multiple regression, AIC seems like fishing for statistical significance to the detriment of clarity about what is actually being tested. Many authors erroneously interpret AIC analyses by interpreting the removal of a factor from the “best” model as meaning that this factor is not statistically/biologically significant. This is faulty logic as the unincluded factor’s statistical significance has not been evaluated if it is not included in the model. The authors seem to have good theoretical reasons for the design as carried out and there don’t seem to be an excess of variables – why not just test the full model?

Specific Comments

lines 38-39: What exactly is “social enforcement”? Is this a new piece of jargon?

line 72: I don’t believe that Kevin’s 2008 paper had anything to do with social interactions or “feedback” (what exactly do you mean by that word anyway?).

line 74: “predicted” not “hypothesized”

line 78: “hypothesized” not “predicted”

---

## Round 0.2 · accepted · Accept

I appreciated the thorough response to the reviewers comments. Your careful explanations about what can and can't be interpreted from the statistics on the age-specific calling curves made sense, and underlines why you cannot provide interpretations with statistical confidence. I also greatly appreciated your explanation about why you removed individuals that did not call in the first 13 days. Knowing that these individuals NEVER called makes it self-explanatory as to why they had to be excluded from your analyses. Lastly, your explanation for why you did not include days prior to the first mating treatment was helpful. Overall, I feel you have done an admirable job on this manuscript, and I look forward to seeing it come out in print.